# Extracts from Cabbage Leaves: Preliminary Results towards a “Universal” Highly-Performant Antibacterial and Antifungal Natural Mixture

**DOI:** 10.3390/biology11071080

**Published:** 2022-07-20

**Authors:** Aldo Arrais, Fabio Testori, Roberta Calligari, Valentina Gianotti, Maddalena Roncoli, Alice Caramaschi, Valeria Todeschini, Nadia Massa, Elisa Bona

**Affiliations:** 1Dipartimento per lo Sviluppo Sostenibile e la Transizione Ecologica, Università del Piemonte Orientale, 13100 Vercelli, Italy; 20025403@studenti.uniupo.it (F.T.); 20014048@studenti.uniupo.it (R.C.); valentina.gianotti@uniupo.it (V.G.); alice.caramaschi@uniupo.it (A.C.); 2Dipartimento di Scienze e Innovazione Tecnologica, Università del Piemonte Orientale, 15121 Alessandria, Italy; maddalena.roncoli@uniupo.it (M.R.); valeria.todeschini@uniupo.it (V.T.); nadia.massa@uniupo.it (N.M.)

**Keywords:** savoy cabbage, white cabbage, diethyl ether extraction, antibacterial activity, antifungal activity, *C. albicans*, *C. glabrata*, *S. aureus*, *P. aeruginosa*, *K. pneumoniae*

## Abstract

**Simple Summary:**

The large antibiotic consumption in the clinical, veterinary, and agricultural fields has resulted in a tremendous flow of antibiotics into the environment. This has led to enormous selective pressures driving the evolution of antimicrobial resistance in bacteria and yeasts. For this reason, the World Health Organization is promoting research to discover new natural products competitive with synthetic drugs in clinical performances. Compared with conventional drugs, the production of natural pharmaceuticals often has a lower environmental impact and lower economic costs of processes, especially when they originate from agricultural wastes. In the context of a circular economy, we aimed to successfully present preliminary results for the valorization of agricultural waste produced in cabbage cultivation by isolating a highly performant antibacterial and antifungal lipophilic natural mixture from cabbage leaves.

**Abstract:**

As dramatically experienced in the recent world pandemic, viral, bacterial, fungal pathogens constitute very serious concerns in the global context of human health. Regarding this issue, the World Health Organization has promoted research studies that aim to develop new strategies using natural products. Although they are often competitive with synthetic pharmaceuticales in clinical performance, they lack their critical drawbacks, i.e., the environmental impact and the high economic costs of processing. In this paper, the isolation of a highly performant antibacterial and antifungal lipophilic natural mixture from leaves of savoy and white cabbages is proposed as successful preliminary results for the valorization of agricultural waste produced in cabbage cultivation. The fraction was chemically extracted from vegetables with diethyl ether and tested against two *Candida* species, as well as *Pseudomonas aeruginosa*, *Klebsiella pneumoniae* and *Staphylococcus aureus* reference strains. All the different fractions (active and not active) were chemically characterized by vibrational FT-IR spectroscopy and GC-MS analyses. The extracts showed high growth-inhibition performance on pathogens, thus demonstrating strong application potential. We think that this work, despite being at a preliminary stage, is very promising, both from pharmaceutical and industrial points of view, and can be proposed as a proof of concept for the recovery of agricultural production wastes.

## 1. Introduction

The World Health Organization (WHO) encourages research aimed to develop medical features using natural products. The excessive consumption of antibiotics in the clinical, veterinary, and agricultural fields has resulted in a huge input of antibiotics into the environment. This has led to selective pressures driving the evolution of antimicrobial resistance (AMR) in commensal and pathogenic microorganisms. AMR is an expensive problem, considering both the human health and social costs. To combat AMR, a Global Action Plan was released in 2015 by the WHO. As part of this plan, six bacterial species were identified, for which the discovery and development of new drugs is required [1]. The acronym ESKAPE encloses these human pathogens, typically associated with nosocomial infections: *Enterococcus faecium*, *Staphylococcus aureus*, *Klebsiella pneumoniae*, *Acinetobacter baumanni*, *Pseudomonas aeruginosa*, and *Enterobacter* species [2,3]. The microorganisms classified under ESKAPE encompass both Gram-positive and -negative bacterial species, and share the capability to “escape” the biocidal effects of antibiotics. Although there has been a reduction in the percentage of methicillin-resistant *S. aureus* (MRSA) strains, MRSA continues to be an important pathogen in European countries, also showing combined resistance to another antimicrobial group. In addition to bacteria, yeasts such as *Candida* spp. cause common infections, which can affect mucous membranes, but can also become systemic [4]. While most *Candida* infections are caused by *C. albicans*, other species including *C. glabrata* and *C. auris*, are frequently resistant and more deadly [5]. Therapies against yeast are mainly based on azoles and polyenes. 

Cabbages (*Brassica oleracea* L.), belonging to the family *Brassicaceae* (Burnett) or *Cruciferae*, are a widely distributed and cultivated vegetable. They originate from the south and western coast of Europe. The Brassicaceae contain a large amount of different healthy compounds such as vitamins (A, C, K, and B6), carotenoids, polyphenols (chlorogenic and sinapic acid derivatives, and flavonoids), minerals (selenium, potassium, and manganese), and nitrogen-sulfur derivatives (glucosinolates and isothiocyanates) [6,7,8,9,10]. It is well-known from the literature that many of these molecules have anticarcinogenic, antimicrobial, anti-inflammatory, and antidiabetic activities [9,10,11,12]. 

The antibacterial activity of different kind of extracts from Brassicaceae plant organs are reported in the literature. For example, Pacheco-Cano et al. [13] demonstrated the biological activity of extracts from the flowers and stems of broccoli cv. Avenger. Their inhibitory effect was proved against pathogenic bacteria (*Bacillus cereus*, *Staphylococcus xylosus*, *S. aureus*, *Shigella flexneri*, *Shigella sonnei* and *Proteus vulgaris*), phytopathogenic fungi (*Colletotrichum gloeosporioides* and *Asperigillus niger*), and yeasts (*C. albicans* and *Rhodotorula* sp.) [13]. Moreover, Andini studied the antibacterial activity of different compounds from Brassicaceae [14] against bacteria, such as *Helicobacter pylori*, *Escherichia coli*, *Bacillus cereus*, *B. subtilis*, *Listeria monocytogenes*, *S. aureus* and yeasts such as *C. albicans*. Finally, Jaiswal et al. tested extracts from fresh Irish Brassica vegetables (York cabbage, white cabbage, broccoli and Brussels sprouts) against Gram-negative (*Salmonella abony* and *Pseudomonas aeruginosa*) and Gram-positive (*L. monocytogenes* and *Enterococcus faecalis*) bacteria [12].

In the context of a circular economy, in this work, we aimed at developing a method for the valorization of agricultural waste produced in the cultivation of cabbage. We isolated a highly performant antibacterial and antifungal lipophilic natural mixture. The new potential antimicrobial extracts were evaluated against the above-mentioned bacteria and yeasts, proposed by the Global Action Plan.

## 2. Materials and Methods

### 2.1. Extract Production

#### 2.1.1. Materials

White and savoy cabbages were purchased in Novara (Piedmont, Italy), in different locations and temporal seasons (spring 2020–autumn and winter 2021). Diethyl ether was purchased by Sigma-Aldrich (Merk Life Science S.r.l., Milan, Italy). Deionized water was obtained through an ion-exchange resin.

#### 2.1.2. Extraction Procedures

The extraction procedure and mixture characterization are summarized in Figure 1.

White cabbage (cultivar certified by the producer) (1.4 kg) was exfoliated, and the leaves were pressed at the bottom of a 5 L round glass bottle, provided with a closure tip, and then soaked under 1.5 L of fresh diethyl ether. The bottle was firmly tipped, after having ascertained that every single leaf was covered by the organic solvent, and left standing for over 7 days, to allow full chemical extraction of the cabbage lipophilic components. Next, the bottle was opened, and the liquid was transferred into large beakers. During the organic extraction, two distinct phases were observed and separated with a separatory funnel: a pale-yellow upper one, containing the diethyl ether solvent with extracted cabbage lipophilic formulations, and a denser orange lower one. The solution in the upper fraction was dried in a rotavapor (40 °C, under reduced pressure), yielding about 500 mg, corresponding on average to 0.06 weight percent of the total cabbage mass. This was then stored in a freezer (−28 °C) prior to chemical analyses and biological antibacterial and antifungal assays. Upon residual solvent evaporation under a ventilated hood, the denser, lower fraction provided a highly viscous, dark-orange, honey-like formulation, extracted from vegetables after a severe long-term organic solvent extraction. Similarly, this formulation was stored in freezer prior to analyses and biological tests.

Savoy cabbage (cultivar certified by the producer) was processed with a closely related extraction protocol, yielding parallel results. Chemical extractions were maintained by soaking up to 10 days, without relevant differences in the observed extracted results.

### 2.2. Antifungal and Antibacterial Assays

The agar disc diffusion method was employed to determine the antifungal and antibacterial activity of the cabbage extracts, according to the methods previously published [15,16,17,18]. The extract was provided as powder and suspended in 1,4 dioxane (Sigma-Aldrich, St. Louis, MO, USA) at a concentration of 0.60 mg µL^−1^ that was considered the starting solution for the following assays.

The antifungal assays were carried out with *C. albicans* ATCC 14053 and *C. glabrata* ATCC 15126. The antifungal effects of clotrimazole (10 μg) and extracts were evaluated according to the M44-A method proposed by the Clinical and Laboratory Standards Institute Standard (CLSI). Clotrimazole (Biolife, Italy) (10 μg) discs were used as positive control. 1,4 dioxane (Sigma-Aldrich, St. Louis, MO, USA; 10 μL) and discs were used as negative control. Plates were incubated at 37 °C for 48 h. All experiments were performed in triplicate. The sensitivity test for the extract was considered positive if it resulted in an inhibition halo higher than that induced by clotrimazole (positive control ≥100%). 

The antibacterial assays were performed with *Staphylococcus aureus* NCTC6571, *Pseudomonas aeruginosa* ATCC27853 and *Klebsiella pneumoniae* ATCC13883 reference strains. Vancomycin (Biolife, Italy), imipenem (Biolife, Italy) and meropenem (Biolife, Italy) effects were evaluated according to EUCAST disk diffusion method for antimicrobial susceptibility v. 7.0. 1,4 dioxane (10 μL) disks were used as negative controls, while vancomycin, meropenem and imipenem were considered as positive control. Plates were incubated at 37 °C for 24 h. All experiments were performed in triplicate. The halos were measured in mm using calipers. The extract was considered active when it produced a halo equal to or higher than positive control (positive control ≥100%).

Moreover, the minimal inhibitory concentration (MIC) of the extracted mixture against bacterial and yeast reference strains was measured according to the EUCAST antifungal MIC method for yeasts (EUCAST definitive document E.DEF 7.3) and EUCAST reference method for bacteria with some modifications performed by the authors and previously published [15,16,17,18]. All microtiter plates were incubated at 37 °C for 24 h. Each experiment was repeated three times. 

### 2.3. FT-IR Analyses

Fourier-transform infrared (FT-IR) measurements were performed by a Thermo-Fisher Scientific Nicolet iSC50 spectrophotometer (Thermo Fisher Scientific, Waltham, MA, USA).

The analyses were performed in transmission mode on the dryed cabbage extracts by the classical method normally employed for solid samples. 

In more detail, 3 mg of the solid sample was pestled with 100 mg of KBr (FT-IR grade) in an agate mortar using a rotatory movements until a homogeneus and impalpable mixture was obtained. Then, the mixture was transferred into a tablet press and pressed at 10 tons cm^−2^, obtaining a 1 cm-diameter disc which was gently transferred into the sample holder of the FT-IR and analysed.

Spectra were collected in transmission mode averaging 100 scans performing a scan at 2 cm^−1^ resolution in the wavelength region from 400 to 4000 cm^−1^.

### 2.4. GC-MS

The chromatographic characterization was performed using a Finnigan Trace GC-Ultra and Trace DSQ. In particular, the gas chromatographic separation was performed using a capillary column Phenomenex ZB-WAX (30 m length, 0.25 mm I.D., and 0.25 μm film thickness). The inlet temperature was set to 250 °C in splitless mode, and helium was used as the carrier gas with a constant flow of 1.0 mL min^−1^. The initial oven temperature was set to 45 °C and reached 250 °C with the ramps reported in Table 1.

The mass spectrometer (MS) transfer line temperature was set to 290 °C. The MS signal was acquired through El+ mode with an ionization energy of 70.0 eV and a source temperature of 290 °C. The solvent delay was set to 6.50 min. The detection was carried out in full-scan mode in the range of 35–500 *m*/*z*.

The samples were prepared by dissolving the cabbage extracts in dichloromethane (50 mg in 1.0 mL) and filtrating by PTFE membrane filters of 0.20 μm porosity. After a dilution of 1:5 in dichloromethane, the extracts were injected in the GC.

### 2.5. Statistical Analysis

The disk diffusion results were statistically analyzed using one-way ANOVA followed by Tukey’s HSD multiple comparisons of means using R (v. 3.5.1) (R Core Team, 2020). Data are presented as boxplots. Differences were considered significant for *p*-values < 0.05.

## 3. Results and Discussion

The white cabbages and savoy cabbages purchased from the local market were extracted by a solvent extraction and a liquid–liquid purification approach without the aid of temperature and pressure to avoid any kind of degradation of the molecules extracted. The extracts were extensively characterized using FT-IR and GC-MS analyses. Meanwhile, the biological activity was evaluated and quantified by antifungal and antibacterial assays together with the determination of minimal inhibitory concentration (MIC).

### 3.1. Biological Activity

The results from the antifungal assays, obtained by the disk diffusion method, are presented in Figure 2.

In more detail, both the two tested extracts demonstrated an antifungal activity statistically higher than clotrimazole (positive control and reference antifungal drugs for candidiasis) against *C. albicans* (Figure 2A), while against *C. glabrata*, only the extract from the white cabbage showed an activity like the antifungal drug (Figure 2B).

The disk diffusion assay results against Gram-positive and Gram-negative bacterial strains are presented in Figure 3. The white cabbage extract showed a statistically higher inhibition effect against *S. aureus* (Gram-positive bacteria) in respect to vancomycin, while the savoy cabbage extract induced an inhibition effect comparable to the reference drug. *P. aeruginosa* and *K. pneumoniae*, both Gram-negative bacteria, showed a similar growth inhibition, lower than the positive control, in the presence of the two cabbage extracts. Nevertheless, the halo size indicated the effectiveness of these two extracts in the reduction of bacterial growth, suggesting their potential use as adjuvants in the antibiotic treatment. 

The MIC results obtained by the standard microdilution method are shown in Table 2. The MIC results confirmed a higher inhibitory effect against both the two *Candida* species and Gram-positive bacteria (e.g., *S. aureus*). Moreover, these extracts showed a lower, but effective, inhibition activity against Gram-negative bacteria. These results demonstrated the “universal” or “comprehensive” potential use of these cabbage extracts. The results obtained against *P. aeruginosa* and *K. pneumoniae*, even if not statistically significant, are very promising because of the difficulty in finding molecules of natural origin that have minimal efficacy against these microorganisms carrying multiple antibiotic resistances. Therefore, even a minimal effectiveness of different extracts allows us to think of mixtures that can have an efficacy similar to that of effective antibiotics such as meropenem (an antibiotic that is currently very effective).

### 3.2. FT-IR

FT-IR is a useful non-destructive method for the preliminary characterization of the molecular structure of different chemical compounds present in plant extracts [19].

Even if from such complex mixtures, which are the natural extracts that contain many molecules, a punctual identification of all the components is not possible; the measurement is useful to gain a first piece of evidence of having extracted the desired components.

FT-IR spectra of formulations extracted from savoy cabbage (blue pattern) and white cabbage (red pattern) are reported in Figure 4. In particular, in Figure 4A, the results related to the biologically active fraction of the obtained mixture are presented, while in Figure 4B, spectral data are reported for the recovered inactive byproducts. In Figure 4A, the two spectra indeed show a closely similar profile, which suggests a related composition extracted from the savoy and white cabbages. Considering that cabbages of different varieties produced in different years and in different production sites were processed, the spectral profile obtained, almost overlapping, guarantees that the extraction method, although unconventional, yields a biologically active fraction in a highly reproducible way.

In more detail, in the biologically active fractions under examination (Figure 4A), the expected lipophilic nature of the formulations can be confirmed from the abundant aliphatic moieties (hereafter, identified by GC-MS analyses, referred to as manifold structures, among which are tetratetracontane and octacosane), providing strong C-H stretching peaks in the 3000–2750 cm^−1^ spectral range, accompanied by different C-H bending modes (normally mixed with other hydrocarbon C-C skeletal modes) in the 1500–1250 cm^−1^ region which can be observed. Moreover, the intense broad band at 3448 cm^−1^ contains O-H stretching modes that can be ascribed to molecules with alcoholic functions (as 1-hydroxy-4-methyl-2,6-di-tert-butylbenzene (BHT) identified by GC-MS).

In the 1800–1650 cm^−1^ spectral range, intense indented IR profiles account for different organic C=O carbonyl groups (ascribed, for example, to myristaldehyde, 2-hexadecanone and methyl tridecyl ketone in both cabbages), whereas broad patterns in the 1200–1000 cm^−1^ range can be related to different C-O stretching modes of ether moieties (e.g., β-sitosterol in white cabbage and BHT in savoy cabbage). Of note, the two diagnostic sharp, strong peaks in the 800–650 cm^−1^ spectral range can be related to out-of-plane gamma (C-H) bending modes of aromatic hydrogens (among the most intense IR vibrations in the entire spectral pattern) [20,21].

### 3.3. Cabbage Extract GC-MS Characterization

Both the white and savoy cabbage extract samples were analyzed by GC-MS in the conditions detailed in the experimental section.

A typical chromatogram obtained by GC-MS is reported in Figure 5. The extracts from the two cabbages contain several volatile (molecules present in the first part of the chromatograms at low retention time values) and semivolatile molecules (molecules with higher retention time). The chromatographic peaks’ identification was performed by a comparison of the spectra recorded with the Wiley spectra library with the criterion of almost 40% matching probability.

The identified compounds are reported in Table 3, together with semiquantative data of the amount measured in the samples. In fact, the peak areas’ values are reported. The comparison of such values is possible since the weights and the analytical procedure performed on the samples are identical both for white and savoy cabbage.

The results of the peak identification showed a higher complexity of the white cabbage (WCE) compared to the savoy cabbage extract (VCE); this complexity could explain the higher biological performance in growth inhibition. Different compounds, with demonstrated antimicrobial activity, were exclusively present in the WCE: 4-(methylsulfanyl) butanenitrile, 1-tridecanal, 1-isothiocyanato-3-(methylthio)-propane, S-methylmethanethiosulphonate, myristic acid, β-sitosterol, 15-nonacosanone, 17-pentatriacontene and 1,2-hexadecanediol. Eight different compounds were present in both cabbage extracts (highlighted in grey in Table 3): dimethyltrisulfide, 5-methyl-1,3-thiazole, 2-hexadecanone, myristaldehyde, methyl tridecyl ketone, 3-(tert-Butyl)-3,4-dihydro-2H-1,4-benzoxazine, tetratetracontane and octacosane. Four of these are known in the literature for their antimicrobial activity. Dimethyltrisulfide is a sulfur compound reported in different cabbage species. 2-Hexadecanone, present both in white and savoy cabbage, was also identified in the essential oil of *Aquilaria crassna*, a tropical plant from south-east Asia and New Guinea belonging to the Thymelaeaceae family. The powder from this plant is used as incense, and the essential oil demonstrates antibacterial activity against *S. aureus* and *C. albicans* [56]. This compound is also one of the main components of essential oil from two species of Fabaceae, *Cassia fistula*, growing in Egypt [57], and *Senna podocarpa*, from different central–north West African countries [58]. In detail, Adebayo et al. demonstrated the antibacterial activity of the essential oil from *S. podocarpa* against *Bacillus subtilis*, *S. aureus*, *Escherichia coli*, *Kiebsiella* spp., *Proteus* spp., *Pseudomonas* spp., *Salmonella* spp., *Penicillium notatum* and *Rhizopus stolonifer*. Myristaldehyde, a component of *Sagittaria trifolia* and *Cinnamomum loureiroi* essential oil, has antimicrobial and antibacterial activities, as reported in Table 3. Moreover, 5-methyl-1,3-thiazole, present in both extracts and other molecular derivates, has also demonstrated antimicrobial activity. Furthermore, from the point of view of the amount of compounds extracted, the results indicated that white cabbage is characterized by a greater number of peaks with a higher quantity present (the peak area is almost always greater than white cabbage).

## 4. Conclusions

Extracts from savoy and white cabbages were obtained using a solid–liquid extraction procedure by diethyl ether and successfully tested for their antibacterial and antifungal properties. In fact, they perform well against two *Candida* species and a *Pseudomonas aeruginosa*, *Klebsiella pneumoniae* and *Staphylococcus aureus* reference-strain blend.

The proposed approach is positioned as a feasibility study with respect to exploiting agricultural waste, easily available and grown anywhere in the world.

In light of the properties shown by the extracts, they will be able to find applications as treatment for mucosal infections from *Candida* strains and for *S. aureus* skin infections. They were less effective against Gram-negative bacteria, even if the demonstrated inhibitory effect must be further investigated. These results are of interest regardless, considering the difficulty in finding natural mixtures that have minimal efficacy against these microorganisms similar to that of effective antibiotics such as meropenem (an antibiotic that is currently very effective).

Given the preliminary nature of the work, the continuation will go in different directions. On the one hand, the antimicrobial potential of other varieties, not only of cabbage but also of other popular vegetables, will be evaluated. From the point of view of the procedure, we will evaluate the scalability from the laboratory to the industrial scale.

Finally, the authors will not only compare the ethyl ether extraction method used in the present work with more conventional extractions—such as ethyl acetate for flavone extraction and n-hexane for the essential oil—but will also test new green solvents (eutectic, deep eutectic, and ionic liquid) in order to obtain a better extraction yield.

## Figures and Tables

**Figure 1 biology-11-01080-f001:**
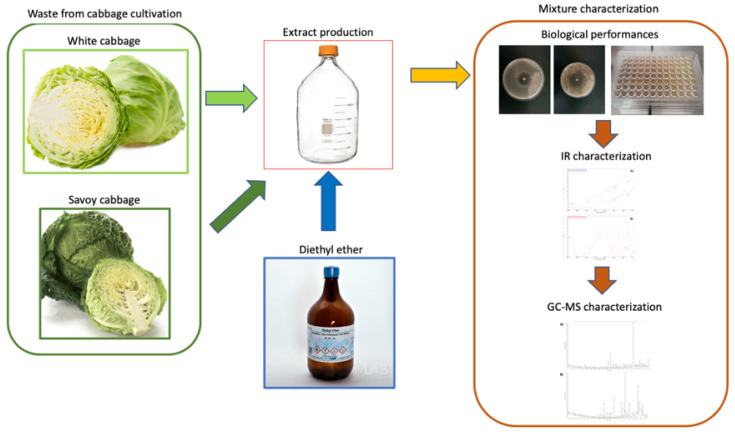
Extraction procedure; biological and chemical characterization of the produced mixture.

**Figure 2 biology-11-01080-f002:**
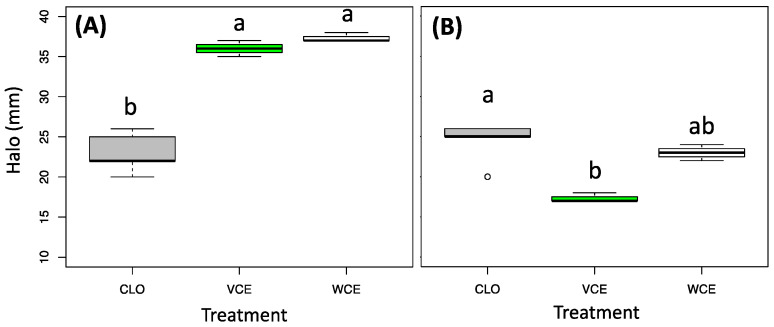
Antifungal assay results, using the disk diffusion method (halo diameter (mm)) from extracts obtained from Savoy cabbage (green box, VCE) and white cabbage (white box, WCE) induced in *C. albicans* ATCC 14053 (**A**) and *C. glabrata* ATCC 15126 (**B**) compared with Clotrimazole (Grey-CLO). Different letters above the bars indicate significant differences, according to Kruskal-Wallis followed by Nemenyi’s post hoc test (*p*-value cutoff = 0.05).

**Figure 3 biology-11-01080-f003:**
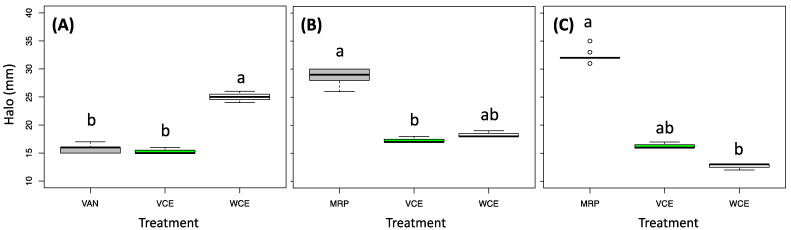
Antibacterial assay results using the disk diffusion method (halo diameter (mm)) from extracts obtained from Savoy cabbage (green box, VCE) and white cabbage (white box, WCE) induced in *S. aureus* NCTC6571 (**A**), *P. aeruginosa* ATCC27853 (**B**), and *K. pneumoniae* ATCC13883 (**C**) compared with the reference antibiotic drug (grey): vancomycin (VAN) and meropenem (MRP). Different letters above the bars indicate significant differences, according to Kruskal–Wallis followed by Nemenyi’s post-hoc test (*p*-value cutoff = 0.05).

**Figure 4 biology-11-01080-f004:**
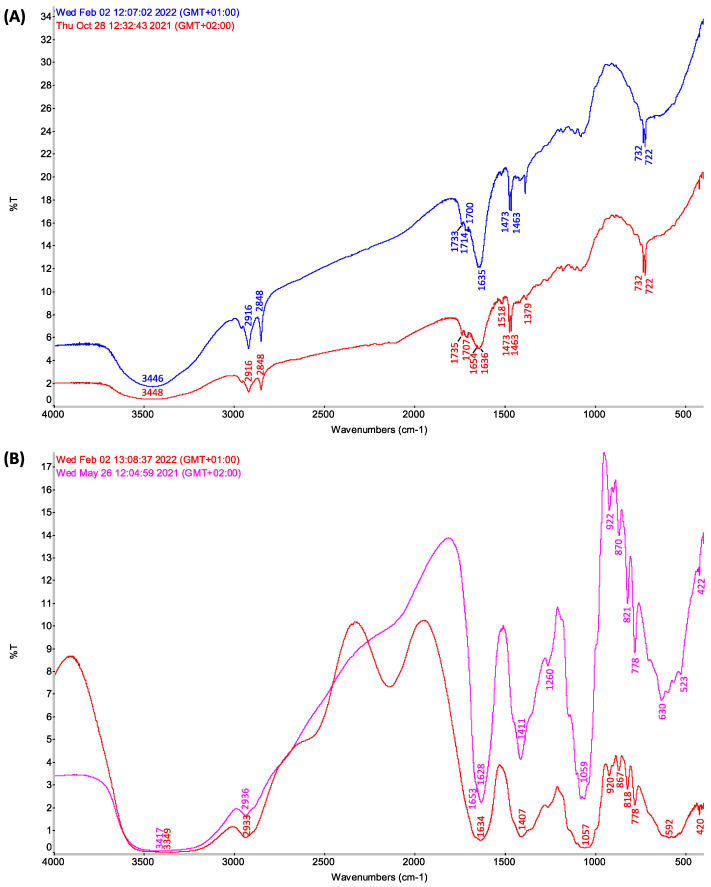
(**A**) Comparison of the IR profiles of the extracts featured with biological activity obtained using diethyl ether from Savoy cabbage (blue line) and white cabbage (red line). Experimental observed peaks (in cm^−1^): savoy cabbage, 3450 (vs, vbr), 3013 (vw), 2960 (s), 2920 (vs), 2851 (vs), 1737 (s), 1712 (br), 1641 (vs, br), 1471 (vs, sh), 1465 (vs, sh), 1414 (w, br), 1382 (vs, sh), 1299 (vs, br),1263 (br), 1198, 1175 (s), 1135, 1111 (s), 1078 (vs), 989 (w), 965, 922, 892 (s), 861 (w), 821 (w), 780 (w, br), 746 (w, br), 732 (vs, vsh), 722 (vs, vsh), 560 (br); white cabbage, 3450 (vs, vbr), 3013 (vw), 2960 (s), 2920 (vs), 2851 (vs), 1737 (s), 1712 (br), 1641 (vs, br), 1471 (vs, sh), 1465 (vs, sh), 1414 (w, br), 1380 (s, br), 1299 (vs, br), 1263 (br), 1198, 1175 (s), 1135, 111 (s), 1078 (vs), 989 (w), 965, 922, 892 (s), 861 (w), 821 (w), 780 (w, br), 746 (w, br), 732 (vs, vsh), 722 (vs, vsh), and 560 (br). (**B**) Comparison of the IR profiles of the sulfured honey-like by-products, extracted with diethyl ether, without biological activity, from Savoy cabbage (red line) and white cabbage (purple line). Experimental observed peaks (in cm^−1^): Savoy cabbage, 3400 (vs, vbr), 2938 (vs, br), 2885 (s, br), 1670 (s, br), 1639 (vs, br), 1457 (br), 1412 (vs, br), 1348 (br), 1263 (s, br), 1184 (br), 1145 (br), 1100 (br), 1078 (br), 1060 (br), 1032 (br), 987 (br), 922 (vs, br), 898 (br), 870 (vs, br), 821 (vs, br), 782 (vs, br), 703 (br), 635 (vs, br), 594 (vs, br), 562 (vs, br), 525 (vs, br), 422 (s, br); white cabbage, 3400 (vs, vbr), 2938 (vs, br), 2885 (s, br), 1670 (s, br), 1639 (vs, br), 1457 (br), 1410 (vs, br), 1348 (br), 1263 (s, br), 1184 (br), 1145 (br), 1100 (br), 1078 (br), 1060 (br), 1032 (br), 987 (br), 922 (vs, br), 898 (br), 870 (vs, br), 821 (vs, br), 782 (vs, br), 703 (br), 635 (vs, br), 594 (vs, br), 562 (vs, br), 525 (vs, br), 422 (s, br). Legend: s strong, vs very strong, w weak, br broad, vbr very broad, sh sharp, vsh very sharp.

**Figure 5 biology-11-01080-f005:**
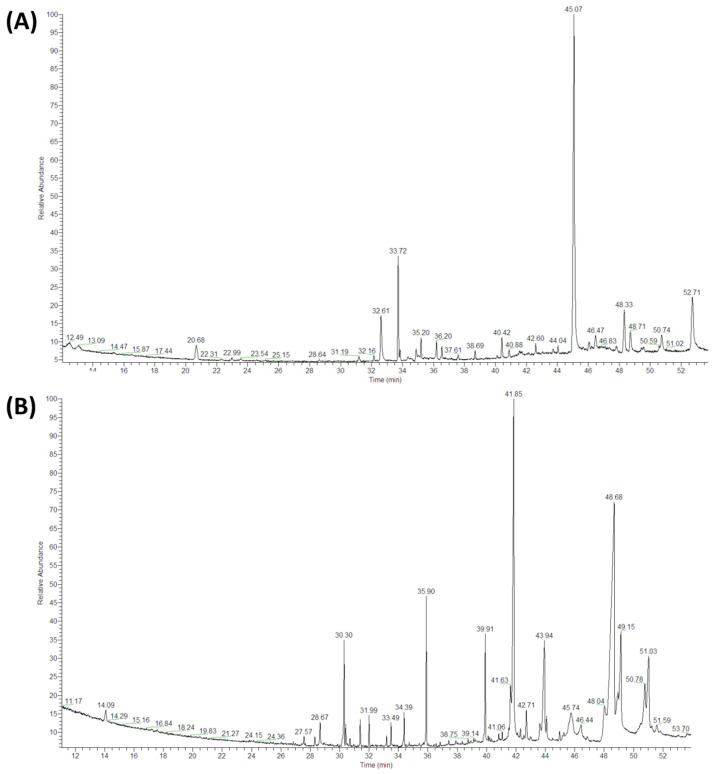
GC-MS chromatograms of diethyl-ether extracts from Savoy (**A**) and white (**B**) cabbage leaves. The chromatographic characterization was performed using a Finnigan Trace GC-Ultra and Trace DSQ. The inlet temperature was set to 250 °C in splitless mode, and helium was used as the carrier gas with a constant flow of 1.0 mL/min.

**Table 1 biology-11-01080-t001:** Oven temperature program.

	Rate (°C/min)	Temperature (°C)	Hold Time (min)
Initial		45.0	2.0
Ramp 1	3.0	100.0	0.1
Ramp 2	5.0	135.0	0.1
Ramp 3	8.0	250.0	12.0

**Table 2 biology-11-01080-t002:** Minimal inhibitory concentration (MIC) obtained by the microdilution method using extracts from Savoy and white cabbage.

Microorganisms ^1^	Savoy Cabbage (VCE)	White Cabbage (WCE)
*C. albicans* ATCC14053	0.062%	0.062%
*C. glabrata* ATCC15126	0.062%	0.062%
*S. aureus* NCTC16571	0.125%	0.062%
*P. aeruginosa* ATCC27853	0.25%	>4%
*K. pneumoniae* ATCC13883	0.25%	>4%

VCE: Savoy cabbage extracts; WCE: White cabbage extracts. ^1^ Reference strains used for the biological activity assays.

**Table 3 biology-11-01080-t003:** Compounds identified by GC-MS method.

CAS n°	Retention Time (min)	Compound	Chemical Class	Savoy Cabbage Peak Area/10^5^	White Cabbage Peak Area/10^5^	Info	Ref.
**541-58-2**	17.2	2,4-Dimethyl -thiazole	Aromatic heterocycle	0	88	-----------	--------
**3658-80-8**	20.7	Dimethyl trisulfide	Thioether	154	612	Antimicrobial activity	[22]
**59121-24-3**	27.5	4-(Methylsulfanyl) butanenitrile	Nitrile	0	320	Sulfur compound present in *Brassica oleracea* var. gongylodes L. and other Brassicaceae/antimicrobial activity	[14,23]
**10486-19-8**	28.3	1-Tridecanal	Aldehyde	0	284	Component of essential oil from leaves and seeds of *Coriandrum sativum* L./oil from flower of *Retama raetam* (Forssk.)–Antibacterial and antifungal activity	[24,25]
**117461-22-0**	30.3	1,4-Dihydro-9,9-dimethyl-1,4-methanonaphtalene-2,3-dicarbonitrile	0	Dicarbonitrile	75	----------	-----
**121013-28-3**	30.3	Methyl-4,4,7-trimethyl-4,7-dihydroindan-6-carboxylate	Ester	0	75	Component of *Salvia lavandulifolia* Vahl. essential oil/anticancer, antimalaric and anti-inflammatory activity	[26]
**105643-80-9**	30.7	(Allylsulfanyl) acetonitrile	Nitrile	0	222	------------	--------
**289-16-7**	31.2	1,2,4-Trithiolane	Heterocyclic sulfur	34	0	Component of fresh mushroom flavor	[27]
**505-79-3**	31.39	1-Isothiocyanato-3-(methylthio)-propane	Thiocyanate	0	671	Component of *Brassica oleracea* (Kale)tea/Component of Chinese kale (*Brassica oleracea* var. alboglabra) and Green Broccoli 90 (*B. oleracea* Linnaeus var. botrytis Linnaeus)/ Antibacterial and anticancer activity	[28,29,30]
**2949-92-0**	31.40	S-methylmethanethiosulfonate	Thiosulfonate	0	663	Component of cabbage (*Brassica* sp.) extracts/Component of the essential oil from *Scorodophloeus zenkeri/* Antimicrobial and antifungal activity	[22,31,32]
**112-54-9**	32.16	Lauraldehyde	Aldehyde	26	0	Component of essential oil from *Eryngium triquetrum* (Apiaceae)/Antimicrobial activity	[33]
**3581-89-3**	32.61	5-Methyl-1,3-thiazole	Aromatic heterocycle	387	901	Antibacterial activity	[34]
**18787-63-8**	33.49	2-Hexadecanone	Ketone	62	447	Component of essential oil from *Cochlospermum tinctorium* A. Rich, *Zingiber officinale* Roscoe, *Salvia anatolica*/Anticancer and antiplasmodial activity	[35,36,37,38]
**114087-07-9**	33.58	[1-(Mercaptomethyl)-2-propenyl] carbamic acid methyl ester	Ester	0	19	------------	-----------
**128-37-0**	33.72	1-Hydroxy-4-methyl-2,6-di-tert-butylbenzene (BHT)	Alkylated phenol	435	98	Antioxidant activity	[39]
**124-25-4**	33.83	Myristaldehyde	Aldehyde	35	607	Component of *Sagittaria trifolia* and *Cinnamomum loureiroi* essential oil/Antimicrobial and antibacterial activities	[40,41]
**116228-46-7**	34.39	2-Methyl-4-(1H-pyrazol-4-YL)- 3-butyn-2-ol	Aromatic heterocycle	0	637	-------------	-----------
**56666-96-7**	34.87	N,N-Dimethyl-N,N-dimethoxysulfinyl-hydrazine	Hydrazine	58	0	------------	------------
**2345-28-0**	35.20	Methyl tridecyl ketone	Ketone	77	795	Component of essential oil from *Epilobium parviflorum* Schreb	[42]
**10396-80-2**	36.20	2,6-Di(t-butyl)-4-hydroxy-4-methyl-2,5-cyclohexadien-1-one	Ketone	89	0	-------------	-----------
**32278-16-3**	38.69	3-(tert-Butyl)-3,4-dihydro-2H-1,4-benzoxazine	Heterocyclic bicycle	39	0	-------------	----------
**544-63-8**	39.91	Myristic acid	Saturated long-chain fatty acid	0	3220	Extracts from leaves and seeds of *Brassica juncea*/Component of seeds from *Myristica fragrans* and *Anethum graveolens;* component of essential oil from leaves and flower of *Helicrisum pallasii* (Streng)/ antibacterial activity	[43,44,45,46]
**18756-03-1**	40.42	(E)-1-Azido-2- phenylethene	Azide	101	0	------------	-----------
**614-96-0**	40.90	5-Methyl-1H-indole	Aromatic heterocycle	39	0	-----------	-----------
**103-23-1**	41.06	Bis(2-ethylhexyl) adipate	Ester	0	205	-----------	------------
**133778-59-3**	42.31	2,3-Dihydro-1,3-methano-1H-cyclopenta[B]quinoxaline	Aromatic heterocycle	0	376	-----------	-----------
**1731-88-0**	42.60	Methyl tridecanoate	Ester	44	0	Extracts from *Acacia pennata* Willd./biological activity as drug against Alzheimer’s disease	[47]
**7098-22-8**	43.85	Tetratetracontane	Long chain alkane	321	6984	Antioxidant and cytoprotective activities	[48]
**629-80-1**	44.98	Hexadecanal	Aldehyde	0	386	------------	------------
**630-02-4**	45.07	Octacosane	Straight-chain alkane	2565	15,364	Antimicrobial activity	[49]
**83-46-5**	46.33	β-Sitosterol	Sterol	0	1219	Antimicrobial, anticancer, anti-inflammatory, anti-asthma, diuretic antiarthritic.	[39,49]
**2764-73-0**	48.68	15-Nonacosanone	Ketone	0	30,507	Antimicrobial activity	[49]
**57-11-4**	48.71	Octadecanoic acid	Carboxylic acid	138	0	Component of *Azadirachta indica* A.Juss (Neem) leaves/ antifungal, antitumor and antibacterial activities	[50]
**60-33-3**	50.74	(Z,Z)-9,12-Octadecadienoic acid	Carboxylic acid	112	0	Extracts from *Gossypium barbadense* seeds; component of extracts from *Brachystegia eurycoma*/antimicrobial activity	[51,52]
**6971-40-0**	50.78	17-Pentatriacontene	Alkene	0	2189	Antinflammatory, anticancer, antibacterial, antiarthritic	[53]
**6920-24-7**	51.03	1,2-Hexadecanediol	Diol	0	4718	Extracts from *Curcuma aromatica* and *Coscinium fenestratum*/Antibacterial and antifungal activity	[54]
**77899-03-7**	51.59	1-Heneicosyl formate	Ester	0	567	Biocontrol activity	[55]
**463-40-1**	52.72	Linolenic acid		593	0	------	--

## Data Availability

Not applicable.

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
