# Peer review of "Extracts from Cabbage Leaves: Preliminary Results towards a “Universal” Highly-Performant Antibacterial and Antifungal Natural Mixture"

_biology, 2022, doi:10.3390/biology11071080_

Round 1
Reviewer 1 Report
In this work the authors reported the isolation of highly-performant antibacterial and antifungal lipophilic natural mixture from savoy and withe cabbages.
According to authors conclusion this work could be very promising, both from a pharmaceutical and industrial point of view, since they propose the re-use of agri-cultural production waste from cabbage.
This reviewer feel that the study is well organized and written, with a potential pharmaceutical application, but some questions need to be answered.
Firstly, the authors use leave cabbages extracts. This is not well specified in the title, abstract and in the main manuscript. This can cause misunderstanding in the reader. I suggest to better specify this aspect.
Even though the cabbage is commonly consumed by population, cytotoxicity of leave cabbage extracts need to be analysed. I suggest to test the cytotoxicity of the extract against human cell cultures.
To add more importance to the study I suggest to test the synergistic action of the leave cabbage extracts with antibiotics against the gram negative pathogens. This data could improve the results and respond to the question raised by the authors…line 234 effectiveness of these two extracts in the reduction of bacterial growth, suggesting their potential use as adjuvant in the antibiotic treatment.
Minor:
Line 128: “antibacterial activity of the different fungal extract”…is this sentence correct? Why did the authors write fungal extracts?
Line 150: “1,4 Dioxane (Sigma- Aldrich,…….” and “organic linseed oil (10 µl) discs”…..please specify why the authors use these negative control. Maybe, are these the solvents?
Line 308-309: “These compounds could be used as “universal” treatment for mucosal infections from Candida strains”. I think that this sentence is too much strong since the activity against C. glabrata is not so interesting as for C. albicans. Please, revise the conclusions.
Author Response
Responses to reviewers
Responses to Reviewer 1
In this work the authors reported the isolation of highly-performant antibacterial and antifungal lipophilic natural mixture from savoy and white cabbages. According to authors conclusion this work could be very promising, both from a pharmaceutical and industrial point of view since they propose the re-use of agricultural production waste from cabbage. This reviewer feel that the study is well organized and written, with a potential pharmaceutical application, but some questions need to be answered.
The authors thank the reviewer for the positive comments.
Firstly, the authors use leave cabbages extracts. This is not well specified in the title, abstract and in the main manuscript. This can cause misunderstanding in the reader. I suggest to better specify this aspect.
The authors thank the reviewer for his/her suggestions and modified the title, abstract and mail text.
Even though the cabbage is commonly consumed by population, cytotoxicity of leave cabbage extracts need to be analysed. I suggest to test the cytotoxicity of the extract against human cell cultures.
The authors thank the reviewer for the suggestion. We will perform this assay in the next future and we will insert the results in the next full paper.
To add more importance to the study I suggest to test the synergistic action of the leave cabbage extracts with antibiotics against the gram negative pathogens. This data could improve the results and respond to the question raised by the authors…line 234 effectiveness of these two extracts in the reduction of bacterial growth, suggesting their potential use as adjuvant in the antibiotic treatment.
The authors thank the reviewer for the suggestion. We will perform this assay in the next future and we will insert the results in the next full paper.
Minor:
Line 128: “antibacterial activity of the different fungal extract”…is this sentence correct? Why did the authors write fungal extracts?
The authors are very sorry but this is a mistake. The authors correct it in the text.
Line 150: “1,4 Dioxane (Sigma- Aldrich,…….” and “organic linseed oil (10 µl) discs”…..please specify why the authors use these negative control. Maybe, are these the solvents?
Yes, they are solvents used to resuspend the dried extracts.
Line 308-309: “These compounds could be used as “universal” treatment for mucosal infections from Candida strains”. I think that this sentence is too much strong since the activity against C. glabrata is not so interesting as for C. albicans. Please, revise the conclusions.
The authors thank the reviewers and correct the text accordingly.
Reviewer 2 Report
Regarding the Manuscript entitled "A "universal" highly-performant antibacterial and antifungal natural mixture, chemically extracted from cabbages" I have a few comments:
1. The Manuscript need extensive English revision by professional or at least by a native speaker.
2. The sentence in lines 16-20 in Simple Summary are repeated in lines 24-28 in Abstract. I recommend to re-write it.
3. line 29: should be "savoy and white cabbages", please correct it.
4. The name "Candida" should be italic, e.g. line 31 is without italic, please improve in whole manuscript.
5. line 58 the citation is not in proper way according to Biology Template, please improve it. Similar in lines 62 and 171.
6. There is not enough news in article. Only two extracts savoy and white cabbages were tested, extraction was performed only with one solvent (diethyl ether). It would be interesting to perform extraction with other solvents e.g. ethyl acetate, acetone and comparison the content of compounds.
7. line 139: "Dioxane", should be 1,4-dioxane, the same as in line 150.
8. In the Materials and Methods chapter, there is no information about the statistics of results.
9. Figure 2: No x-axis caption in two graphs.
10. Table 2. MIC should be presented in concentration microgram/mililiter no in %.
11. Only FT-IR and GC-MS analysis were performed to detect compounds in extracts. It would be interesting to isolate the main substances from extracts and compare the biologicial activites of extracts with pure compounds.
Author Response
Responses to reviewers
Responses to Reviewer 2
Regarding the Manuscript entitled "A "universal" highly-performant antibacterial and antifungal natural mixture, chemically extracted from cabbages" I have a few comments:
1. The Manuscript need extensive English revision by professional or at least by a native speaker.
The authors thank the reviewers and correct the text accordingly.
- The sentence in lines 16-20 in Simple Summary are repeated in lines 24-28 in Abstract. I recommend to re-write it.
The authors thank the reviewers and correct the text accordingly.
line 29: should be "savoy and white cabbages", please correct it.
The authors thank the reviewers and correct the text accordingly.
The name "Candida" should be italic, e.g. line 31 is without italic, please improve in whole manuscript.
The authors thank the reviewers and correct the text accordingly.
line 58 the citation is not in proper way according to Biology Template, please improve it. Similar in lines 62 and 171.
6. There is not enough news in article. Only two extracts savoy and white cabbages were tested, extraction was performed only with one solvent (diethyl ether). It would be interesting to perform extraction with other solvents e.g. ethyl acetate, acetone and comparison the content of compounds.
The authors, as suggested by the editor, decided to modify the work as “communication” in order to present the preliminary results and to complete the work using also other kind of solvents.
line 139: "Dioxane", should be 1,4-dioxane, the same as in line 150.
The authors thank the reviewers and correct the text accordingly.
In the Materials and Methods chapter, there is no information about the statistics of results.
The authors thank the reviewers and correct the text accordingly adding the following sentence..
“The disk diffusion results were statistically analyzed using one-way ANOVA followed by Tukey’s HSD multiple comparisons of means using R (v. 3.5.1). Data are presented as boxplots. Differences were considered significant for p-values < 0.05. “
Figure 2: No x-axis caption in two graphs.
The description of the variable measured is indicated in the graph.
Table 2. MIC should be presented in concentration microgram/mililiter no in %.
Because of the starting material is liquid the authors think that is more correct to use % as measurement unit.
Only FT-IR and GC-MS analysis were performed to detect compounds in extracts. It would be interesting to isolate the main substances from extracts and compare the biologicial activites of extracts with pure compounds.
The authors thank the reviewer and will do this analysis in the future and present the results obtained in the next complete paper.
Reviewer 3 Report
The manuscript titled "A "universal" highly-performant antibacterial and antifungal natural mixture, chemically extracted from cabbages" by Arrais and co-workers described the evaluation of the antibacterial and antifungal activity of diethyl ether extracts from savoy and white cabbages.
In my opinion the manuscript is well written and reported promising data, but it resulted to be too preliminary for the pubblication. In this form, the manuscript is better suitable for a medicinal chemistry/ natural products journal.
Several issues should be addressed by the authors before the publication in Biology journal:
-Biological evaluation: I think this work represents a preliminary investigation about the potential antibacterial activity of the extracts, but not enough for the publication in a Q1-biology (JCR) journal. The authors should perform further biological tests in order to assess the antibacterial activity of the extracts and investigate the potential mechanism of action. Moreover, since the authors discussed about multi-drug resistant bacteria, they should evaluate their extracts against clinical isolates of resistant species.
-Introduction: the introduction provides a sufficient background about the topic, but it seems to be too focused on the drug resistant microorganisms, since the authors didn't investigate the antibacterial activity of extracts against resistant species.
-Methods: the chemical characterization methods of the extracts (GC-MS and FT-IR) should be located after the Extraction procedures
-Results: in order to keep the same section order, the "chemical part" should be located before the biological tests, as in the "method" section
-Results: I think that a brief discussion about the extraction process (before the characterization experiments) could be useful for the readers.
Some other minor issues:
-line 29. "withe" is "white"
-line 31 "candida" shoul be in italics
-line 52 "antidrug resistant" is "drug resistant"
-Table 2. the table 2 lacks details and footnote
Author Response
Responses to reviewers
Responses to Reviewer 3
The manuscript titled "A "universal" highly-performant antibacterial and antifungal natural mixture, chemically extracted from cabbages" by Arrais and co-workers described the evaluation of the antibacterial and antifungal activity of diethyl ether extracts from savoy and white cabbages. In my opinion the manuscript is well written and reported promising data, but it resulted to be too preliminary for the publication. In this form, the manuscript is better suitable for a medicinal chemistry/ natural products journal.
The authors thank the reviewer for the comments and they are in agreement with His/her opinion. In fact the authors, in agreement with the editor, changed the publication as “communication” in order to present preliminary data.
Several issues should be addressed by the authors before the publication in Biology journal:
-Biological evaluation: I think this work represents a preliminary investigation about the potential antibacterial activity of the extracts, but not enough for the publication in a Q1-biology (JCR) journal. The authors should perform further biological tests in order to assess the antibacterial activity of the extracts and investigate the potential mechanism of action. Moreover, since the authors discussed about multi-drug resistant bacteria, they should evaluate their extracts against clinical isolates of resistant species.
The authors thank the reviewer and will do this analysis in the future and present the results obtained in the next complete paper.
-Introduction: the introduction provides a sufficient background about the topic, but it seems to be too focused on the drug resistant microorganisms, since the authors didn't investigate the antibacterial activity of extracts against resistant species.
The authors thank the reviewer and modify the text.
-Methods: the chemical characterization methods of the extracts (GC-MS and FT-IR) should be located after the Extraction procedures.
The authors thank the reviewer and will do this analysis in the future and present the results obtained in the next complete paper.
-Results: in order to keep the same section order, the "chemical part" should be located before the biological tests, as in the "method" section
The authors thank the reviewer and will do this analysis in the future and present the results obtained in the next complete paper.
-Results: I think that a brief discussion about the extraction process (before the characterization experiments) could be useful for the readers.
Some other minor issues:
-line 29. "withe" is "white"
Done
-line 31 "candida" shoul be in italics
Done
-line 52 "antidrug resistant" is "drug resistant"
Done
-Table 2. the table 2 lacks details and footnote
Done
Reviewer 4 Report
The manuscript " A "universal" highly-performant antibacterial and antifungal natural mixture, chemically extracted from cabbages" fits the journal's scope. The design of the research is simple, the methodology well-known, and correct. The authors present their results regarding the antifungal and antibacterial activity of a lipophilic extract from two species of cabbage: white cabbage and Savoy cabbage. The chemical composition of the extracts was evaluated by GC-MS, and its biological properties, using disc difussion and MIC methods.
Before publication the manuscript needs some corrections:
1. Please indicate the scientific name and the variety of each cabbage sample.
2. Lines 133-135: please indicate the yield of extraction for Savoy cabbage
3. Please mention in the manuscript how was establish the identity of each variety
4. Please indicate if voucher specimens are available
5. Please indicate the origin of bacterial and fungal strains
6. Please justify the use of FT/IR analysis.
Author Response
Responses to reviewers
Responses to Reviewer 4
The manuscript " A "universal" highly-performant antibacterial and antifungal natural mixture, chemically extracted from cabbages" fits the journal's scope. The design of the research is simple, the methodology well-known, and correct. The authors present their results regarding the antifungal and antibacterial activity of a lipophilic extract from two species of cabbage: white cabbage and Savoy cabbage. The chemical composition of the extracts was evaluated by GC-MS, and its biological properties, using disc diffusion and MIC methods.
Before publication the manuscript needs some corrections:
1. Please indicate the scientific name and the variety of each cabbage sample.
Done
Lines 133-135: please indicate the yield of extraction for Savoy cabbage.
Please mention in the manuscript how was establish the identity of each variety
Done
Please indicate if voucher specimens are available
Please indicate the origin of bacterial and fungal strains.
The origin of the different reference strains was ATCC.
Please justify the use of FT/IR analysis.
This techniques was choosen to evaluate the characteristics of each extracts.
Reviewer 5 Report
1. Authors should avoid vogue terms; what is mean by universal and highly performant. Simple solvent extracts were tested against bacterial pathogens.
2. Bacterial and fungal sources, their standardization and susceptibility pattern against common antibiotics is missing.
3. Chemical analysis via GC-MS is incomplete. Authors are advised to add calculated
retention index (Kovat´s index) and compare with those reported in the literature to confirm the tentative identification of the compounds. For confirmation of the identification, a co-injection of the chemical standards of the tentatively identified compounds must be incorporated.
4. GC-MS chromatogram ar of low resolution
5. FTIR has no role in compounds identification here
6. Authors antimicrobial discovery is of limited significance than they claimed in the title; Discussion and conclusion are poorly presented.
7. Need careful language changes
Author Response
Responses to reviewers
Responses to Reviewer 5
- Authors should avoid vogue terms; what is mean by universal and highly performant. Simple solvent extracts were tested against bacterial pathogens.
The authors thank the reviewer and will do this analysis in the future and present the results obtained in the next complete paper.
- Bacterial and fungal sources, their standardization and susceptibility pattern against common antibiotics is missing.
- Chemical analysis via GC-MS is incomplete. Authors are advised to add calculated
retention index (Kovat´s index) and compare with those reported in the literature to confirm the tentative identification of the compounds. For confirmation of the identification, a co-injection of the chemical standards of the tentatively identified compounds must be incorporated.
The authors thank the reviewer and will do this analysis in the future and present the results obtained in the next complete paper.
- GC-MS chromatogram ar of low resolution.
Done
- FTIR has no role in compounds identification here
- Authors antimicrobial discovery is of limited significance than they claimed in the title; Discussion and conclusion are poorly presented.
The authors thank the reviewer.
- Need careful language changes.
Done
Round 2
Reviewer 1 Report
The manuscript has been improved.
Accepted in the present form.
Author Response
Thank you for your support and positive opinion.
Reviewer 2 Report
Authors improved manuscript and now it is suitable to published as Communication.
Author Response
Thank you
Reviewer 3 Report
The revisions done from the authors are sufficient for the pubblication as “communication” , since the authors reported preliminary data
Author Response
Thank you
Reviewer 5 Report
1. In which solvent was extract dissolved and how it was applied is still not clear
2. The susceptibility pattern and standardization of tested strains is still not clear to standard antibiotics; kindly refer to this article; this might help in refining the methods. https://link.springer.com/article/10.1186/s12906-016-1491-4;
3. The manuscript is too preliminary regarding its antimicrobial and phytochemical analysis. I suggested to calculate Kovat index rather than simple retention time.
4. I am still not convinced with the terms highly performant and universal. What is the extract antimicrobial comparison with standard. Is it really highly performant?
5. Your results figures are not clear, provide the results in different form like table. DIZ ar any other results are not easy for reader to understand.
6. with a huge background in the abstract, your study is very limited with no detailed analysis and poor phytochemical analysis. The mansucript still need a lot of work to be published in 5 plus impact factor
7. There is no correlation of he identified compounds with the current antimicrobial results
Author Response
Reviewer 5
Comments and Suggestions for Authors
- In which solvent was extract dissolved and how it was applied is still not clear.
The biocidal fractions were extracted with diethyl ether (b.p. 42 ° C, leaves no traces and does not degrade natural substances, in low T treatments); the bio tests have seen the re-dissolution in dioxane (used because it is inactive against the used microorganisms. All this is an unconventional process, but it works: it provides a fraction that works like standard antibiotics, with store-bought cabbage. The authors give this information using the sentence “The extract was provided as powder and suspended in 1,4 Dioxane (Sigma- Aldrich, St. Louis, MO) at the concentration of 0.60 mg µl-1 that was considered the starting solution for the following assays”, lines 121-123.
- The susceptibility pattern and standardization of tested strains is still not clear to standard antibiotics; kindly refer to this article; this might help in refining the methods. https://link.springer.com/article/10.1186/s12906-016-1491-4. The procedure used for standard antibiotics is reported in EUCAST method. This method is produce to standardize all the antibiotic assay in Europe. As you need some more information, please visit the website at www.eucast.org
- The manuscript is too preliminary regarding its antimicrobial and phytochemical analysis. I suggested to calculate Kovat index rather than simple retention time.
The referee is certainly right in suggesting to add the Kovat indices. Unfortunately given the preliminary nature of the work, that we declare throughout the article, we have not set up the analysis to be able to calculate them. Currently, due to GC-MS maintenance problems and availability windows for using the instrument, we are unable to repeat the analysis in the seven days that the editor has granted us for revisions. Moreover, having the detection in mass spectrometry, we have identified and reported in the table only those analytes that, compared with the spectra of the Wiley library both by software automatic algorithm and with the critical observation made by ourselves on the significance of the assignment, have given us “safe” levels of matching.
We agree with the referee that the characterization works on oils or phytoextracts are normally much more extensive, but our intent was quite different. In fact, we focused the work to have a proof of concept about ​​the usability of "poor" foods, such as cabbage, to obtain a natural extract, at low cost and in an eco-sustainable way, that rivals the antibiotics of synthesis in biocidal action, even against strains of pneumonia, a prospect that, for poor or developing countries, can certainly be of scientific interest. Moreover, such focus, in fact, fits perfectly with the meaning and purpose of the Special Issue in which we submitted the article (Food By-Products as Sustainable Sources of Health-Promoting and Anti-microbial Bioactive Molecules).
We probably haven't explained the focus of the article well enough. We certainly add more explanation in the text.
- I am still not convinced with the terms highly performant and universal. What is the extract antimicrobial comparison with standard. Is it really highly performant?
The authors thank the reviewer, but they are confident that these extracts are very promising, and they give the complete information of the chemical characterization in the complete paper. The activity against different strains higher than antibiotics they make these extracts worthy of being reported to the scientific committee also with a view to recovering waste resources.
- Your results figures are not clear, provide the results in different form like table. DIZ ar any other results are not easy for reader to understand.
The authors disagree with the reviewer. The use of box plots is easy to read, as also said by the other reviewers, and more understandable to everyone than a table.
- With a huge background in the abstract, your study is very limited with no detailed analysis and poor phytochemical analysis. The manuscript still need a lot of work to be published in 5 plus impact factor.
The authors are very sorry about the poor opinion of the reviewer about their work. The authors, on the other hand, believe that the work is an excellent starting point to investigate the possibility of enhancing the very abundant waste from cabbage cultivation by producing low-cost antibacterial substances. This aspect is certainly of interest to the scientific community, especially considering the topic of the special issue.
- There is no correlation of he identified compounds with the current antimicrobial results.
In the extracted blend, there are 4 basic components, including BHT which has proven antimicrobial and antifungal action; there is an aliphatic wax, which happens to be also found in essential oils that work (under this or other similar aliphatic forms), which could act as a booster factor, merging with the membranes of microorganisms and thus favoring the action of other substances.
Round 3
Reviewer 5 Report
No further suggestions.
Author Response
The Author thank the reviewer for the suggestions in order to improve you work.
Regards